# Effects of Submaximal Performances on Critical Speed and Power: Uses of an Arbitrary-Unit Method with Different Protocols

**DOI:** 10.3390/sports7060136

**Published:** 2019-05-31

**Authors:** Henry Vandewalle

**Affiliations:** Physiology at Faculty of Medicine, University Paris XIII, 93000 Bobigny, France; henry.vandewalle@club-internet.fr

**Keywords:** critical speed, critical power, performance reliability, modelling

## Abstract

The effects of submaximal performances on critical speed (S_Crit_) and critical power (P_Crit_) were studied in 3 protocols: a constant-speed protocol (protocol 1), a constant-time protocol (protocol 2) and a constant-distance protocol (protocol 3). The effects of submaximal performances on S_Crit_ and P_Crit_ were studied with the results of two theoretical maximal exercises multiplied by coefficients lower or equal to 1 (from 0.8 to 1 for protocol 1; from 0.95 to 1 for protocols 2 and 3): coefficient C_1_ for the shortest exercises and C_2_ for the longest exercises. Arbitrary units were used for exhaustion times (t_lim_), speeds (or power-output in cycling) and distances (or work in cycling). The submaximal-performance effects on S_Crit_ and P_Crit_ were computed from two ranges of t_lim_ (1–4 and 1–7). These effects have been compared for a low-endurance athlete (exponent = 0.8 in the power-law model of Kennelly) and a high-endurance athlete (exponent = 0.95). Unexpectedly, the effects of submaximal performances on S_Crit_ and P_Crit_ are lower in protocol 1. For the 3 protocols, the effects of submaximal performances on S_Crit_, and P_Crit_, are low in many cases and are lower when the range of t_lim_ is longer. The results of the present theoretical study confirm the possibility of the computation of S_Crit_ and P_Crit_ from several submaximal exercises performed in the same session.

## 1. Introduction

### 1.1. Empirical Model of Running and Cycling Performances

Several empirical and descriptive models of performance have been proposed: the power-law model by Kennelly [1], asymptotic hyperbolic models by Hill and Scherrer [2,3], and, more recently, the logarithmic model of Péronnet and Thibault [4] and the 3-parameter asymptotic models by Hopkins [5] and Morton [6]. These empirical models are often used to estimate (i) the improvement in performance (ii) the future performances and running speeds over given distances (iii) the endurance capability, i.e., “the ability to sustain a high fractional utilization of maximal oxygen uptake for a prolonged period of time” [4].

In 1954, Scherrer et al. proposed a linear relationship [3] between the exhaustion time (t_lim_) of a local exercise (flexions or extensions of the elbow or the knee) performed at different constant power outputs (P) and the total amount of work performed at exhaustion (W_lim_) for t_lim_ ranging between 3 and 30 min:W_lim_ = a + bt_lim_ = Pt_lim_

Consequently, the relationship between P and t_lim_ is hyperbolic: a = Pt_lim_ − bt_lim_ = (P − b) t_lim_
t_lim_ = a/(P − b) = a/(P − P_Crit_)
where b is a critical power (P_Crit_).

In 1966, Ettema applied the critical-power concept to world records in running, swimming, cycling, and skating exercises [7] and proposed a linear relationship between the distances (D_lim_) and world records (t_lim_) from 1500 to 10000 m:D_lim_ = a + b_tlim_

It was assumed that the energy cost of running was almost independent of speed (1 kcal.kg^−1^.km^−1^) under 22 km.h^−1^ [8]. Consequently, D_lim_ and parameter “a” were equivalent to amounts of energy. Therefore, parameter “a” has been interpreted as equivalent to an energy store. Thereafter, parameter “a” was considered as an estimation of maximal Anaerobic Distance Capacity (ADC expressed in metres) for running exercises [7,9]. Slope b was considered as a critical velocity (S_Crit_). 

D_lim_ = ADC + S_Crit_ t_lim_

In 1981, the linear W_lim_-t_lim_ relationship was adapted to exercises on a stationary cycle ergometer and it was demonstrated that slope b of the W_lim_-t_lim_ relationship was correlated with the ventilatory threshold [10]. Therefore, slope b was proposed as an index of general endurance (P_Crit_). Thereafter, Whipp et al. [11] proposed another linear model with S_Crit_ (or P_Crit_) and a:S = S_Crit 1/t_ + a (1/t_lim_) Running
P = P_Crit 1/t_ + a (1/t_lim_) Cycling
where S is running speed. 

Actually, the hyperbolic model is often used as it is the simplest model that corresponds to a linear relationship between exhaustion time (t_lim_) and distance (D_lim_) in running and swimming or total work (W_lim_) in cycling. For many exercise physiologists, S_Crit_ and P_Crit_ are considered as fatigue thresholds [12]. Moreover, the values of S_Crit_ and P_Crit_ become endurance indices when they are normalized to Maximal Aerobic Speed (S_Crit_ /MAS) or Maximal Aerobic Power (P_Crit_ /MAP). Parameter “a” is also called ADC (Anaerobic Distance Capacity) or ARC (Anaerobic Running Capacity) in running [7,9] and AWC (Anaerobic Work Capacity) in cycling [13,14]. 

However, the relationship between t_lim_ and D_lim_ is not perfectly linear as suggested by the power-law model by Kennelly:D_lim_ = k t_lim_^g^
S = D_lim_/t_lim_ = (k t_lim_^g^)/t_lim_ = k t_lim_^g−1^
where exponent g can be considered as an endurance index and parameter k is equal to the maximal speed corresponding to the unit of t_lim_ [15]. 

### 1.2. Variability of the Performances of Exhausting Exercises.

Three protocols are used for the estimation of the performances of exhausting exercises:

(1) to run as long as possible at a constant speed or to cycle as long as possible at a constant power output. This constant-speed protocol is often called Time-to-Exhaustion;

(2) to run as much distance (or to produce as much work on a cycle-ergometer) as they can within a given time period (constant-time protocol);

(3) to run a set distance (or to produce a set work on a cycle-ergometer) as fast as possible (constant-distance protocol).

The first protocol (Time-to-Exhaustion) is used for the estimation of S_Crit_ on a treadmill or the estimation of P_Crit_ on a cycle ergometer. The second protocol was used for prediction of one-hour running performance [16]. The third protocol was used in the studies on the modelling of running performances that were based on the world records [17,18,19,20] or performances in the Olympic games [21] or individual performance of elite endurance runners [15]. The reliability of performances in protocol 1 (constant speed) is low, whereas the reliability of the other protocols is higher [22,23,24,25,26,27]. For example in swimming, the Coefficient of Variation of constant-speed protocol (CV = 6.46 ± 6.24%) was significantly less reliable (p < 0.001) than those of constant-time protocol (CV = 0.63 ± 0.54%) and constant-distance protocol (CV = 0.56 ± 0.60%) [27]. 

In a recent study [28] critical speeds measured on a treadmill with a constant-speed protocol were compared with a Single-Visit Field Test of Critical Speed. The constant-speed runs to exhaustion on treadmill were performed with 3 running speeds during 3 separate sessions. Two single-visit field tests on separate days consisted to the measurement of maximal performances over 3600, 2400, and 1200 m (constant-distance protocol) with 30-min or 60-min recovery. Unexpectedly, there was no difference in S_Crit_ measured with the treadmill and 30-min- and 60-min recovery field tests although the reliability of protocol 1 is lower than that of protocol 3. Thereafter, a Single-Visit Field Test of Critical Speed was tested in trained and untrained runners [29]: the reliability of S_Crit_ was better in the trained runners.

### 1.3. Purpose of the Present Study

In few studies, the values of S_Crit_ and P_Crit_ are computed from the best maximal performances of several exhausting exercises of the same subjects [15] or world records [17,18,19,20] or performances in the Olympic games [21]. In most studies, it is not obvious that the data used in the computation of S_Crit_ and P_Crit_ are maximal. For example, the performance variability is important in protocol 1 that is mainly used in laboratories. Moreover, several exhausting exercises are often performed in the same session with protocols 1, 2 and 3, which could increase the performance variability because of fatigue. As suggested in a review [30], the purpose of the present study was to confirm the interest of S_Crit_ and P_Crit_ computed from exercises whose performances are submaximal. 

In the Single-Visit Field Test of Critical Speed, there were trained and untrained runners whose reliability of S_Crit_ was different [29]. Therefore, the effects of submaximal performance on S_Crit_ and P_Crit_ in the present study have been compared between a low-endurance athlete and a high-endurance athlete, i.e., athletes with low and high endurance indices (for example, exponent g or S_Crit_/MAS or P_Crit_/MAP). The values of exponent g were about 0.95 in the best elite endurance runners [15,31] as Gebrselassie whose ratio S_Crit_/MAS was equal to 0.945 (MAS corresponded to the maximal running speed during 7 min). In the low-endurance runners whose ratios S_Crit_/MAS were equal to 0.764 ± 0.078, exponent g was about 0.80 [31].

As S_Crit_ and P_Crit_ can be computed from two exhausting exercises, the effects of submaximal performances on these indices have been estimated by multiplying both theoretical maximal data by two coefficients: C_1_ for the shortest performances and C_2_ for the longest performances. 

The effects of submaximal performances on S_Crit_ and P_Crit_ were estimated with arbitrary units for t_lim_, D_lim_ and S (or P) in the present study. Indeed, there are many different cases: the range of t_lim_ can be different for each athlete in protocols 1 and 3;the range of speeds and distances (or power-output and work in cycling) can be different for each athlete in protocol 2;the same endurance indices can correspond to different maximal aerobic speed (MAS) or maximal aerobic power (MAP in cycling).

The effects of submaximal performances on endurance indices have been tested for values of t_lim_ equal to 1 and 4 (arbitrary units), which corresponds to the usual range of t_lim_ (3–12m in [28,29]) in many study. In some studies, the range of t_lim_ is 2–15 min [12,32,33], Therefore, the effects of submaximal performances have also been tested for values of t_lim_ equal to 1–7 (arbitrary unit). 

## 2. Methods

### 2.1. Arbitrary Units

The values of t_lim1_, D_lim1_ and S_1_ in arbitrary units in protocols 1, 2 and 3 are equal to 1:t_lim1_ = 1   and   S_1_ = 1   and   D_lim1_ = 1

#### 2.1.1. Arbitrary Units in Protocols 1 and 2 

The values of t_lim2_ and t_lim3_ in arbitrary units in protocols 1 and 2 are equal to 4 and 7, respectively.

t_lim2_ = 4   and   t_lim3_ = 7

The distances and running speeds corresponding to t_lim2_ and t_lim3_ was computed from the power-law model b Kennelly with arbitrary units of t_lim_: D_lim1_ = k t_lim1_^g^ = k (1^g^) = 1 then k = 1   and   D_lim_ = t_lim_^g^ and   S = t_lim_^g−1^

For the high-endurance athlete, exponent g of the power-law model is equal to 0.95. Therefore_,_ the values of D_lim2_, D_lim3_, S_2_ and S_3_ in arbitrary units are equal to: D_lim2_ = t_lim2_^g^ = 4^0.95^ = 3.7321   and   D_lim3_ = t_lim3_^g^ = 7^0.95^ = 6.3510

S_2_ = t_lim2_^g−1^ = 4^−0.05^ = 0.9330   and   S_3_ = t_lim3_^g−1^ = 7^−0.05^ = 0.9073

For the low-endurance athlete, exponent g of the power-law model is equal to 0.80. Therefore_,_ the values of D_lim2_, D_lim3_, S_2_ and S_3_ in arbitrary units are equal to:D_lim2_ = 4^0.80^ = 3.0314   and   D_lim3_ = 7^0.80^ = 4.7433

S_2_ = 4^−0.2^ = 0.7579   and   S_3_ = 7^−0.2^ = 0.6776

#### 2.1.2. Arbitrary Units in Protocol 3 

The values of constant-distances (D_lim_) in the present study were equal to the averages of the distances of low and high-endurance athletes, in protocols 1 and 2, i.e., 1 (D_lim1_), 3.3819 (D_lim2_) and 5.5471 (D_lim3_). The values of t_lim2_ and t_lim3_ corresponding to these distances were:D_lim_ = t_lim_^g^

(t_lim_^g^)^1/g^ = t_lim_ = D_lim_^1/g^

t_lim1_ = D_lim1_^1/g^ = 1^1/g^ = 1   and   t_lim2_ = D_lim2_^1/g^ = 3.3819^1/g^and t_lim3_ = D_lim3_^1/g^ = 5.5471^1/g^

For the low-endurance athlete:t_lim2_ = 3.3819^1/0.8^ = 4.5862    and   t_lim3_ = 5.5471^1/0.80^ = 8.5130

For the high-endurance athlete:t_lim2_ = 3.3819^1/0.95^ = 3.6059   and   t_lim3_ = 5.5471^1/0.95^ = 6. 0705 

### 2.2. Coefficients C_1_ and C_2_


The ranges of coefficients C_1_ and C_2_ used in protocol 2 and 3 were from 0.95 to 1. Indeed, in protocols 2 and 3, the submaximal performances are the results of submaximal speeds (or submaximal power outputs in cycling). If the ratio C_1_/C_2_ is lower than 0.9330, the speed corresponding to t_lim2_ would be higher than the speed at t_lim1_ in the high-endurance athlete. In protocol 1 (constant-speed protocol), the speed (or power) does not depend on C_1_ or C_2_ and the variability of performances are higher [27]. Therefore, the ranges of C_1_ and C_2_ were larger (from 0.8 to 1). 

### 2.3. Computation of the Effects of Submaximal Performances on S_Crit_ and P_Crit_


#### 2.3.1. Computation in the Constant-Speed (or Constant Power Output) Protocol (Protocol 1)

The submaximal performances are the result of the submaximal values of t_lim_. The submaximal values of t_lim_ (t_lim1 sub_ and t_lim2 sub_) are:t_lim1 sub_ = C_1_ t_lim1_   and   t_lim2 sub_ = C_2_ t_lim2_

Therefore, the submaximal values of D_lim_ (D_lim1 sub_ and D_lim2 sub_) are equal to:D_lim1 sub_ = S_1_ t_lim1 sub_ = S_1_C_1_ t_lim1_ = C_1_ D_lim1_

D_lim2 sub_ = S_2_ t_lim2 sub_ = S_2_C_2_ t_lim2_ = C_2_ t_lim2_

For running:S_Crit_ = (D_lim2_ − D_lim1_)/(t_lim2_ − t_lim1_)

S_Crit sub_ = (C_2_ D_lim2_ − C_1_ D_lim1_)/(C_2_ t_lim2_ − C_1_ t_lim1_)

S_Crit sub_/S_Crit_ = [(C_2_ D_lim2_ − C_1_ D_lim1_)/(C_2_ t_lim2_ − C_1_ t_lim1_)]/[(D_lim2_ − D_lim1_)/(t_lim2_ − t_lim1_)] (1)

For cycling: P_Crit sub_/P_Crit_ = [(C_2_ W_lim2_ − C_1_ W_lim1_)/(C_2_ t_lim2_ − C_1_ t_lim1_)]/[(W_lim2_ − W_lim1_)/(t_lim2_ − t_lim1_)] 

#### 2.3.2. Computation in the constant-time protocol (protocol 2)

The submaximal performances are the result of submaximal speeds.

S_1sub_ = C_1_S_1_    and    S_2sub_ = C_2_S_2_

D_lim1 sub_ = C_1_S_1_ t_lim1_ = C_1_D_lim1_    and    D_lim2 sub_ = C_2_S_2_ t_lim2_ = C_2_D_lim2_

For cycling, the submaximal performances correspond to lower powers.

P_1sub_ = C_1_P_1_    and    P_2sub_ = P_2_S_2_

W_lim1 sub_ = C_1_P_1_ t_lim1_ = C_1_ W_lim1_    and    W_lim2 sub_ = C_2_P_2_ t_lim2_ = C_2_ W_lim2_

For running:S_Crit_ = (D_lim2_ − D_lim1_)/(t_lim2_ − t_lim1_)

S_Crit sub_ = (D_lim2 sub_ − D_lim1sub_)/(t_lim2_ − t_lim1_) = (C_2_ D_lim2_ − C_1_ D_lim1_)/(t_lim2_ − t_lim1_)

S_Crit sub_/S_Crit_ = [(C_2_ D_lim2_ − C_1_ D_lim1_)/(t_lim2_ − t_lim1_)]/[(D_lim2_ − D_lim1_)/(t_lim2_ − t_lim1_)]

S_Crit sub_/S_Crit_ = (C_2_ D_lim2_ − C_1_ D_lim1_) /(D_lim2_ − D_lim1_)(2)

For cycling:P_Crit sub_/P_Crit_ = (C_2_ W_lim2_ − C_1_ W_lim1_)/(W_lim2_ − W_lim1_)

#### 2.3.3. Computation in the constant-distance protocol

The submaximal performances are the result of submaximal speeds. 

S_1sub_ = C_1_S_1_    and    S_2sub_ = C_2_S_2_

For cycling, the submaximal performances correspond to lower powers.

P_1sub_ = C_1_P_1_    and    P_2sub_ = C_2_P_2_

Hence:    t_lim1sub_ = D_lim1_/S_1sub_ = D_lim1_/C_1_S_1_ = t_lim1_/C_1 _

and    t_lim2sub_ = D_lim2_/S_2sub_ = D_lim2_/C_2_S_2_ = t_lim2_/C_2_

For running:S_Crit_ = (D_lim2_ − D_lim1_)/(t_lim2_ − t_lim1_)

S_Crit sub_ = (D_lim2_ − D_lim1_)/(t_lim2sub_ − t_lim1sub_) = (D_lim2_ − D_lim1_)/(t_lim2_/C_2_ − t_lim1_/C_1_)

S_Crit sub_ /S_Crit_ = [(D_lim2_ − D_lim1_)/(t_lim2_/C_2_ − t_lim1_/C_1_)]/[(D_lim2_ − D_lim1_)/(t_lim2_ − t_lim1_)]

S_Crit sub_ /S_Crit_ = (t_lim2_ − t_lim1_)/(t_lim2_/C_2_ − t_lim1_/C_1_)(3)

For cycling:P_Crit sub_ /P_Crit_ = (t_lim2_ − t_lim1_)/(t_lim2_/C_2_ − t_lim1_/C_1_)

## 3. Results

The interest of the use of arbitrary units is demonstrated in Table 1 and Table 2 for protocol 1. Athletes A, B, C, D, E and F who have the same ratio t_lim1_/t_lim2_ (4) and the same ratio S_2_/S_1_ (0.7579) have the same effects (S_Critsub_/S_Crit_) corresponding to the same coefficients C_1_ and C_2_ (Table 1). Moreover, the use of the same arbitrary units can also been applied to cycling exercises (Table 2): the effects (P_Critsub_/P_Crit_) of submaximal performances are the same when ratio t_lim1_/t_lim2_ and ratio P_2_/P_1_ are similar as in running exercises. 

The effects of submaximal performances on endurance indices are presented in Figure 1 (constant-speed protocol), Figure 2 (constant-time protocol) and Figure 3 (constant-distance protocol).

### 3.1. Results for Protocol 1 (Constant Speed or Power Output Protocol)

For protocol 1, five curves of ratio S_Crit sub_/S_Crit_ corresponding to five values of C_1_ (0.80, 0.85, 0.90, 0.95 and 1.00) were computed from equation 1 with an increment of C_2_ equal to 0.001.

The effects of submaximal performances on ratio S_Crit sub_/S_Crit_ (or ratio P_Crit sub_/P_Crit_) are lower in the high-endurance athlete (Figure 1B,D) than in the low-endurance athlete (Figure 1A,C).

In Figure 1B,D, the effects of submaximal performances on ratio S_Crit sub_/S_Crit_ are lower when the range of t_lim_ is longer (1–7 instead of 1–4).

In Figure 1A, the lowest and the highest ratios S_Crit sub_/S_Crit_ are equal to 0.9567 and 1.0295, respectively. 

When C_1_ is equal to C_2_ (empty circles), i.e., when the levels of submaximal performances are the same for both exhausting exercises, there is no effect of submaximal performances on ratio S_Crit sub_/S_Crit_ (or ratio P_Crit sub_/P_Crit_) according to Equation (1):S_Crit sub_/S_Crit_ = [(C_2_ D_lim2_ − C_2_ D_lim1_)/(C_2_ t_lim2_ − C_2_ t_lim1_)]/[(D_lim2_ − D_lim1_)/(t_lim2_ − t_lim1_)] = 1

### 3.2. Results for Protocol 2 (Constant-Time Protocol)

The values of coefficients C_1_ and C_2_ were limited from 0.95 to 1. Six curves of ratio S_Crit sub_/S_Crit_ corresponding to six values of C_1_ (0.95, 0.96, 0.97, 0.98, 0.99 and 1.00) were computed from Equation (2) with an increment of C_2_ equal to 0.001.

As for protocol 1, the effects of submaximal performances on ratio S_Crit sub_/S_Crit_ (or ratio P_Crit sub_/P_Crit_) are lower in the high-endurance athlete (Figure 2B,D) than in the low-endurance athlete (Figure 2A,C). 

In Figure 2C,D, the effects of submaximal performances on ratio S_Crit sub_/S_Crit_ are lower when the range of t_lim_ is longer (1–7 instead of 1–4). 

In Figure 2A, the lowest and the highest ratios S_Crit sub_/S_Crit_ are equal to 0.9254 and 1.0246, respectively.

When C_1_ is equal to C_2_ (empty circles in Figure 2A), i.e., when the levels of submaximal performances are the same for both exhausting exercises, the ratios S_Crit sub_/S_Crit_ (or P_Crit sub_/P_Crit_) are equal to C_2_ (or C_1_) according to Equation (2):S_Crit sub_/S_Crit_ = (C_2_ D_lim2_ − C_2_ D_lim1_)/(D_lim2_ − D_lim1_) = C_2_ (or C_1_)

### 3.3. Results for Protocol 3 (Constant-Distance Protocol)

Six curves of ratio S_Crit sub_/S_Crit_ corresponding to six values of C_1_ (0.95, 0.96, 0.97, 0.98, 0.99 and 1.00) were computed from equation 3 with an increment of C_2_ equal to 0.001.

In contrast with protocols 1 and 2, the effects of submaximal performance on ratio S_Crit sub_/S_Crit_ (or P_Crit sub_/P_Crit_) are more important in the high-endurance athlete. However, in the high-endurance athlete, the ranges of t_lim1_–t_lim2_ (1-3.6059) and t_lim1_–t_lim3_ (1–6.0705) is shorter than the ranges of t_lim1_–t_lim2_ (1–4.5862) and t_lim1_-t_lim3_ (1–8.5130) in the low-endurance athlete.

In Figure 3B, the lowest and the highest ratios S_Crit sub_/S_Crit_ are equal to 0.9321 and 1.0206, respectively.

When C_1_ is equal to C_2_ (empty circles in Figure 3B), i.e., when the levels of submaximal performances are the same for both exhausting exercises, the ratios S_Crit sub_/S_Crit_ (or P_Crit sub_/P_Crit_) are equal to C_2_ (or C_1_) according to Equation (3):S_Crit sub_ /S_Crit_ = (t_lim2_ − t_lim1_)/(t_lim2_/C_2_ − t_lim1_/C_2_) = C_2_ (t_lim2_ − t_lim1_)/(t_lim2_ − t_lim1_) = C_2_ (or C_1_)

## 4. Discussion

The effects of submaximal performances on S_Crit 1/t_ and P_Crit 1/t_ in the model proposed by Whipp et al. [11] are not presented in the present study. Indeed, S_Crit 1/t_ (or P_Crit 1/t_) is equal to S_Crit_ (or P_Crit_) when both indices are computed only from two exhausting exercises with constant-distance [15] or constant-power [34] protocols in running and cycling. Similarly, in the present study, the effects of submaximal performances were the same for S_Crit 1/t_ and S_Crit_ (or P_Crit 1/t_ and P_Crit_) when they were computed from two submaximal exercises whatever the protocol. Consequently, the Figures about the effects of submaximal performances on S_Crit 1/t_ or P_Crit 1/t_ are not added in the present study. 

Previous experimental studies [22,23,24,25,26,27] showed that performance reliability with constant-speed protocol is significantly lower than those with the other protocols (constant-time or constant-distance protocols). However, for S_Crit_ or P_Crit_ in the present theoretical study, the effects of 20%-submaximal performances in protocol 1 are lower than the effects of 5%-submaximal performances in protocols 2 and 3. For example, the lowest ratio S_Crit sub_/S_Crit_ in protocol 1 is equal to 0.9567 (Figure 1A) whereas the lowest ratio in protocol 2 is equal to 0.9254 (Figure 2A). 

Ratio S_Cri tsub_/S_Crit_ (or ratio P_Crit sub_/P_Crit_) is equal to 1 when the maximal and submaximal D_lim_-t_lim_ relationships are parallel i.e., when the distances of both submaximal performances to the maximal D_lim_-t_lim_ line are similar:-C_1_ is equal to C_2_ for protocol 1(empty circles in Figure 1);-D_lim1_ (1 − C_1_) is equal to D_lim2_ (1 − C_2_) for protocol 2_;_-t_lim1_ (1 − 1/C_1_) is equal to t_lim2_ (1 − 1/C_2_) for protocol 3.

In protocol 1, the submaximal performances correspond to similar decreases in D_lim_ and t_lim_ whereas, in protocol 2 and 3, they only correspond to a decrease in D_lim_ or t_lim_ (Figure 4). These simultaneous decreases in D_lim_ and t_lim_ limit the distance between the submaximal performance and the maximal D_lim_-t_lim_ line, which explain the lower effects of submaximal performances on ratio S_Crit sub_/S_Crit_. 

In protocols 2 and 3, when C_1_ is equal to C_2_ (empty circles in Figure 2A, Figure 3B) ratios S_Crit sub_ /S_Crit_ are equal to C_1_ and are not very low (S_Crit sub_/S_Crit_ ≥ 0.95). When C_2_ is lower than C_1_, ratios S_Crit sub_ /S_Crit_ are lower and sometimes not negligible in protocols 2 and 3 (for example, in Figure 2A, S_Crit sub_/S_Crit_ = 0.9254 for C_2_ = 0.95 and C_1_ = 1). However, although the range of C_1_-C_2_ is much larger (0.8–1.0), ratios S_Crit sub_/S_Crit_ are not very low (≥ 0.9567) in protocol 1, even when C_2_ is lower than C_1_ (Figure 1A). 

The effects of submaximal performances on ratio S_Crit sub_/S_Crit_ (or ratio P_Crit sub_/P_Crit_) may be low (S_Crit sub_ /S_Crit_ ≥ 0.95) for both low-endurance and high-endurance athletes in the three protocols. These possible low effects of submaximal performances on ratio S_Crit sub_/S_Crit_ could explain that it is possible to compute S_Crit_ from the values of t_lim_ of 3 trials performed with protocol 3 in a same session with only 30 min of recovery between the trials as in a Single-Visit Field Test [28,29]. This low sensitivity of S_Crit_ or P_Crit_ to submaximal performances was previously suggested in a study on the comparison of critical speeds of continuous and intermittent running exercise on a track [35] and also in a review [30].

The effects of submaximal performances on ratio S_Crit sub_/S_Crit_ are lower in the high-endurance athlete for constant-speed and constant-time protocols (Figure 1 and Figure 2). That said, the effects of submaximal performances on ratio S_Crit sub_/S_Crit_ in constant-distance protocol (protocol 3) are higher in the high-endurance athlete (Figure 3). However, the effects of submaximal performances on ratio S_Crit sub_/S_Crit_ (or ratio P_Crit sub_/P_Crit_) are lower when the range of t_lim_ is longer (for example, 1–7 instead of 1–4) as illustrated in Figure 1, Figure 2 and Figure 3. Therefore, in protocol 3, the shorter ranges of t_lim1_-t_lim2_ and t_lim1_-t_lim3_ in the high-endurance athlete explain these computed higher effects of submaximal performances on ratio S_Crit sub_/S_Crit_. In contrast, the coefficient of variation of the Single-Visit Field Test that corresponds to this constant-distance protocol was lower in trained runners whose S_Crit_ were faster than untrained runners [29]. It was likely that the reliability of S_Crit_ in the trained runners was higher because of control of the maximal running speed corresponding to a given distance and better recovery.

In the Single-Visit Field Test [28], the running performances were submaximal because the values of D’ (equivalent of ADC) were significantly lower in the 30-min (106.4-m) and 60-min-recovery (102.4-m) than in the 3-session treadmill test (249.7 m). However, S_Crit_ in the Single-Visit Field Test was not different of S_Crit_ in the 3-session treadmill test. These results were consistent with those of a previous experimental study on the effects of a 6-min exhausting exercise on S_Crit_ in cycling [36]. In the present theoretical study, the submaximal performances have also effects on parameter “a” (ADC). For example, in protocol 1, parameter “a” decreases when C_1_ and C_2_ are equal and lower than 1 (empty circles in Figure 1) but increases when C_1_ is equal to 1 and C_2_ is lower than 1. These effects of submaximal performances on parameter “a” are not computed in the present study because this parameter is not an endurance index and its meaning is questionable [11].

The present method can also be used for the submaximal-performance effects on the exponent g of the power-law model by Kennelly [1] and the endurance index (EI) of the logarithmic model by Péronnet and Thibault [4] as they are 2-parameter models:D_lim_ = k t_lim_^g^  and   S = k t_lim_^g−1^    power-law model

100 S/MAS = 100 − EI log(t_lim_/t_MAS_)          logarithmic model

## 5. Conclusions

The results of the present theoretical study confirm the interest of S_Crit_ and P_Crit_ computed from exercises whose performances are submaximal and performed in the same session. Indeed, for the 3 protocols, the theoretical effects of submaximal performances on ratio S_Crit sub_/S_Crit_ (or ratio P_Crit sub_/P_Crit_) are low in many cases. The effects of submaximal performances are lower when the ratio t_lim2_ /t_lim1_ is larger. In protocol 3, it is likely that, in practice, the reliability of S_Crit_ is better in trained runners due to the control of the maximal running speed corresponding to a given distance. 

## Figures and Tables

**Figure 1 sports-07-00136-f001:**
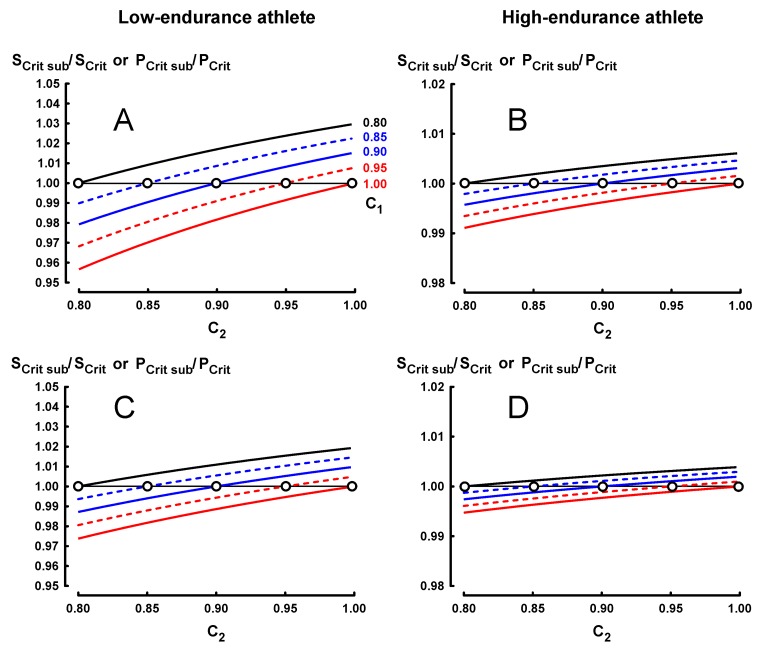
effects of submaximal performances in protocol 1 (constant speed or constant power) on ratios S_Critsub_/S_Crit_ or P_Critsub_/P_Crit_ for the different values of C_1_ and C_2_. Figures **A** and **B** correspond to the range t_lim1_-t_lim2_ (1–4) whereas Figures **C** and **D** correspond to the range t_lim1_-t_lim3_ (1–7). The specifications of the lines are presented in Figure 1A. Empty circles correspond to C_1_ equal to C_2_.

**Figure 2 sports-07-00136-f002:**
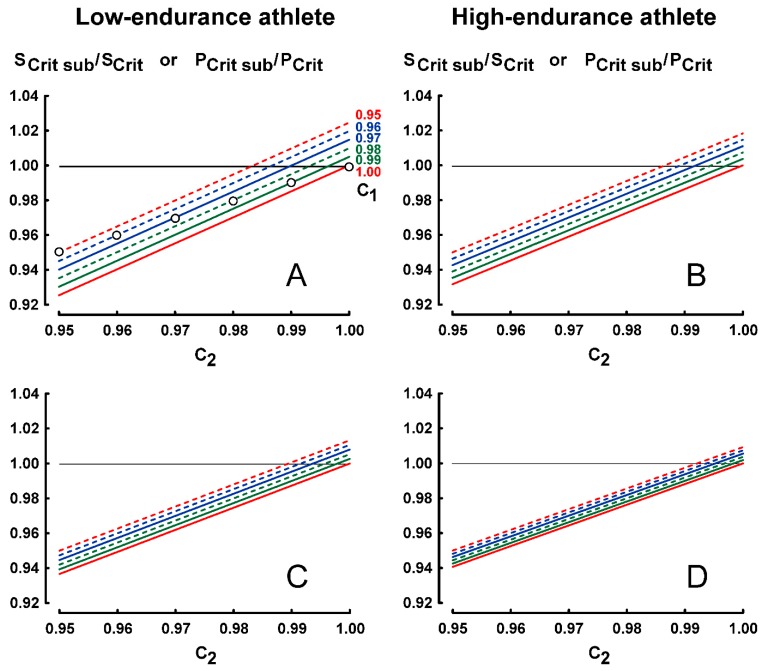
effects of submaximal performances in protocol 2 (constant time-protocol) on ratios S_Crit sub_/S_Crit_ for the different values of C_1_ and C_2_. Figures **A** and **B** correspond to the range t_lim1_-t_lim2_ (1–4) whereas Figures **C** and **D** correspond to the range t_lim1_-t_lim3_ (1–7). The specification of the lines is presented in Figure **A**. Empty circles in Figure **A** correspond to C_1_ equal to C_2_.

**Figure 3 sports-07-00136-f003:**
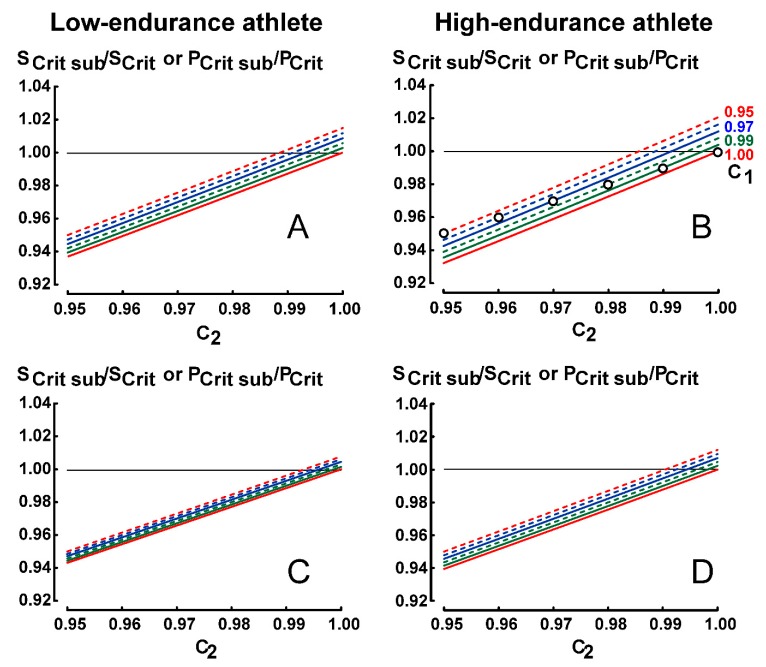
Effects of submaximal performances in protocol 3 (constant distance-protocol) for the different values of C_1_ (specification of the lines in Figure **B** are the same as in Figure 2) and C_2_. Figures **A** and **B** correspond to the range t_lim1_-t_lim2_ whereas Figures **C** and **D** correspond to the range t_lim1_-t_lim3_. Empty circles in Figure **B** correspond to C_1_ equal to C_2_.

**Figure 4 sports-07-00136-f004:**
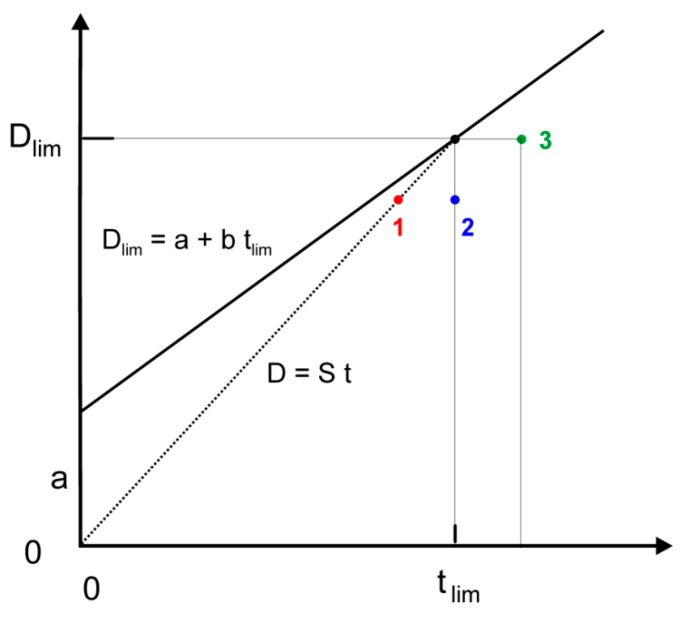
Comparison of the effects of submaximal performances in the 3 protocols. Black dot: maximal performance. Red dot: submaximal performance in protocol 1. Blue dot: submaximal performance in protocol 2. Green dot: submaximal performance in protocol 3. The dotted line corresponds to the relationship between distance (D) and time (t) at a given running speed (S).

**Table 1 sports-07-00136-t001:** Athletes A, B and C have the same values of t_lim1_ and t_lim2_ but different running speeds (S_1_ and S_2_). Athletes D, E and F have the same values of running speed as athletes A, B and C, respectively. The values of t_lim1_ and t_lim2_ are higher in athletes D, E and F but ratio t_lim2_/t_lim1_ is the same and equal to 4.

						C_1_ = 0.9 and C_2_ = 1		C_1_ = 1 and C_2_ = 0.9	
	S_1_	S_2_	t_lim 1_	t_lim 2_	S_Crit_	S_Critsub_	S_Critsub_/S_Crit_	S_Critsub_	S_Critsub_/S_Crit_
	m.s^−1^	m.s^−1^	s	S	m.s^−1^	m.s^−1^		m.s^-1^	
A	4	3.0316	180	720	2.709	2.750	1.015	2.659	0.982
B	5	3.7895	180	720	3.386	3.438	1.015	3.324	0.982
C	6	4.5474	180	720	4.063	4.126	1.015	3.989	0.982
D	4	3.0316	240	960	2.709	2.750	1.015	2.659	0.982
E	5	3.7895	240	960	3.386	3.438	1.015	3.324	0.982
F	6	4.5474	240	960	4.063	4.126	1.015	3.989	0.982

**Table 2 sports-07-00136-t002:** Athletes A, B and C have the same values of t_lim1_ and t_lim2_ but different power-outputs (P_1_ and P_2_). Athletes D, E and F have the same values of power-outputs as athletes A, B and C, respectively. The values of t_lim1_ and t_lim2_ are higher in athletes D, E and F but ratio t_lim2_/t_lim1_ is the same and equal to 4.

						C_1_ = 0.9 and C_2_ = 1		C_1_ = 1 and C_2_ = 0.9	
	P_1_	P_2_	t_lim 1_	t_lim 2_	P_Crit_	P_Critsub_	P_Critsub_/P_Crit_	P_Critsub_	P_Critsub_/P_Crit_
	W	W	S	S	W	W		W	
A	240	182	180	720	163	165	1.015	160	0.982
B	320	243	180	720	217	220	1.015	213	0.982
C	400	303	180	720	271	275	1.015	266	0.982
D	240	182	240	960	163	165	1.015	160	0.982
E	320	243	240	960	217	220	1.015	213	0.982
F	400	303	240	960	271	275	1.015	266	0.982

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
