# Peer review of "Effects of Submaximal Performances on Critical Speed and Power: Uses of an Arbitrary-Unit Method with Different Protocols"

_sports, 2019, doi:10.3390/sports7060136_

Round 1
Reviewer 1 Report
Introduction
L34 - "...the hyperbolic model is currently used..." - insert reference. You mentioned two hyperbolic models (Hill and Scherer). Which one are you referring to?
L37 - 39 - explain to the reader what are "a" and "b" and where these come from.
L40 - 41 - "...not exactly linear as in the other model..." - which model? add reference here. Add reference number after "Kennelly" on line 41.
Line 43 - explain the constant k and g.
Line 67 - 71 - it seems that you suddenly stopped your thoughts. It looks as if you were going to compare different protocols and studies. Single-field test (different distances) vs constant speed protocol, then a single visit of CS in trained and untrained runners. But you never discussed this study to support your arguments from the previous paragraph.
Purpose
You can make your purpose a bit more clear. This is a highly relevant topic, as it looks like you are attempting to establish how submaximal performances / tests can be used to estimate Scrit and Pcrit. Please explain this a bit better and include references when appropriate on lines 74 - 78. Why these coefficients? is this based on past research? Explain / briefly mention again which ones are protocols 1, 2, and 3.
L87 - 90 - when have these been tested for those values (1 and 4, and 1 and 7). Provide references and explain it.
What do you mean by low-endurance and high-endurance athlete? Please provide a clear description of who is classified in these categories. In addition, if you could provide a description of past protocols utilized for protocols 1, 2, and 3 that would enhance the readers understanding of it (perhaps this can be done in the introduction).
Methods
Please explain / make it more clear, and include the appropriate references where needed. You seemed to be more clear in one of your previous papers ("Modelling of running performances: comparisons of power-law, hyperbolic, logarithmic, and exponential models in elite endurance runners") in your description of each model. This paper could benefit from it as well. In the mentioned paper, your description of Kennelly's model is brief, yet good. While it has been described elsewhere, I believe this paper could benefit from such descriptions, considering this is a complex topic.
L110 - 111 - please provide the reference for the exponents. The same is valid for L114-115.
Here or at another point (above section) provide a descriptor for the high- and low-endurance athlete.
Results
This seems a bit confusing too.What do you mean when you are saying that the effect on Scrit and Pcrit are lower in the high endurance athlete? Are you talking about the reliability of the tests to determine their Scrit / Pcrit? Overall, you could improve the clarity of this section.
Discussion
Similar to previous sections, this seems confusing. It looks like the first paragraph could be shortened. At the same time, you could discuss the results in more depth and talk about the applications that these have in for coaches / sport scientists in applied settings while providing other references in the process. You do this a bit right after it, but see notes below.
L261 - 266 - again, what do you mean by the effect is low? And why is this allowing the determination of the Scrit and Pcrit from submaximal tests? What is the connection?
L267 - 272 - this is an important finding - that longer submaximal efforts Tlim 1- 7, 4 to 30min might be more reliable for significance and reliability of Scrit. Still, this is quite a range. Do you have any further recommendations?
Conclusion
Seemed like you just reiterated the results of the study here. What is the application that this has for coaches / sport scientists? As you put previously "3 submaximal tests that can all be performed in a single session can be used to determined Scrit and Pcrit in athletes". This is a significant finding and it is worth reinforcing it here, perhaps providing a recommendation for how it can be used in athletic settings.
Author Response
Thank you for the review.
As suggested by another reviewer, the title of the paper has been changed
“The results of the present theoretical study confirm the possibility of the computation of SCrit and PCrit from several exhausting exercises performed in the same session. “ has been added in lines 20-22.
The references and the description of the hyperbolic model of Scherrer are presented in lines 34-54.
Parameters a and b are explained in lines 41 and 46-50
The power-law model of Kennelly is explained in lines 67-73
A reference of the second protocol is presented in line 84.
The interest of the “Single-Visit Field Test of Critical Speed” in the present study has be developed in lines 91-97 and in lines 107-110 for the interest of low-endurance and high-endurance athletes.
The purpose of the study is presented in lines 99-106
The arbitrary units in the different protocols is presented in lines 126-154
The explanation of the difference of the ranges of C1 and C2 in protocol 1 versus protocols 2 and 3 is presented in lines 155-162.
The legends of tables 1 and 2 have been modified.
The explanations the different curves are presented in lines 226-227, 248-249 and 268-269.
“the effects of submaximal performance on SCrit or PCrit” have been replaced with “the effects of submaximal performance on ratios SCrit sub/SCrit (or PCrit sub/PCrit)”
The interest of the “Single-Visit Field Test of Critical Speed” has been developed in Discussion and Conclusion
Reviewer 2 Report
In this paper, I think that an interesting examination about critical speed and critical power is done.
There is one thing that I want you to improve. Do not span Figure 2 across two pages, because it is difficult to see.
Author Response
Thank you for the rewiew
As suggested by another reviewer, the title of the paper has been changed
The references and the description of the hyperbolic model of Scherrer are presented in lines 34-54.
The power-law model of Kennelly is explained in lines 67-73
The interest of the “Single-Visit Field Test of Critical Speed” in the present study has be developed in lines 91-97 and in lines 99-110 for the interest of low-endurance and high-endurance athletes.
The explanations the different curves are presented in lines 223-224, 243-244 and 263-264.
“the effects of submaximal performance on SCrit or PCrit” have been replaced with “the effects of submaximal performance on ratios SCrit sub/SCrit (or PCrit sub/PCrit)”
The interest of the “Single-Visit Field Test of Critical Speed” has been developed in Discussion and Conclusion
Reviewer 3 Report
General Comments:
- I think the title could be changed to: “Effects of submaximal performances on critical speed and power: Uses an arbitrary-unit method with different protocols “.
- The manuscript included at the end from page 13 below to page 17 the author's guidelines which must be deleted.
Abstract Comments:
- I would suggest more details in the abstract regarding the conclusions whilst represented simply between all investigated protocols.
- Line “9 to 10” and “20 and 21” : I detected two duplicate sentences in the abstract and I prefer to sentences simply and be also short sentences without any duplication.
Introduction Comments:
- As the study proposed to estimate the effects of submaximal performances on the computation of SCrit and PCrit, I observed INSUFFICIENT literature review in the introduction.
- Line 89: “have also be tested” must be “have also been tested”.
Methods Comments:
- Please, introduce shortly at the start of methods, what will be presented regarding the arbitrary units and the computation of the effects of submaximal performances on SCrit and PCrit and for the different three-studied protocol.
- There are few problems with the presentation of the Materials regarding the data analysis or the data collections of the different presented models.
Results Comments:
- Line 186 and 187: The authors refer to a table with numbers 3 and 4, however, no tables included in the whole paper. (The manuscript includes only two tables)
- The results in table (1) and (2) were presented without any header information about variables. Moreover, I would like to understand WHAT are meaning (A, B, C, D, E, F) ???
- There are no term description keys below the tables regarding all abbreviations that stated in both tables.
- Please, rewrite the titles of table (1) and (2) above the tables.
- The authors mentioned to figure number (1) which demonstrate the different values of C1 and C2, however, no curves explain any information about C1. (The authors have presented only the values of C2 as stated in Figure (1) A, B, C, and D). In addition, as same as missed information’s about C1 in Figure (2) and (3)
Discussion Comments:
- The paper misses a serious discussion on the findings of the second and third investigated protocols. The discussion scoped only about the first protocol.
- As claimed in the beginning of the manuscript regarding the main aims of this study, I can’t understand simply in the discussions what this study seeks for; is it a comparison between protocols or a build-up of the new model for the SCrit and PCrit calculations !!??
Conclusions Comments:
- The conclusions seem to be descriptive. It should be rewritten in more details and clearly stating the facts as clearly as possible
Author Response
Thank you for the review.
As suggested, the title of the paper has been changed
The references and the description of the hyperbolic model of Scherrer are presented in lines 34-54.
The power-law model of Kennelly is explained in lines 67-73
The purpose of the study is presented in lines 99-106
The chapter corresponding the arbitrary units for the different protocols in Methods has been improved (lines 126-154) by transfer of data and some missing data have been added.
The explanation of the difference of the ranges of C1 and C2 in protocol 1 versus protocols 2 and 3 is presented in lines 155-162.
The legends of tables 1 and 2 have been modified.
The words “table 3” and “table 4” have been suppressed.
The explanations the different curves are presented in lines 226-227, 248-249 and 268-269.
“the effects of submaximal performance on SCrit or PCrit” have been replaced with “the effects of submaximal performance on ratios SCrit sub/SCrit (or PCrit sub/PCrit)”
The interest of the “Single-Visit Field Test of Critical Speed” has been developed in Discussion and Conclusion
Round 2
Reviewer 1 Report
Abstract
Please modify the abstract according to the changes that were made and make it clearer when arriving at the conclusions.
Introduction
L38 - 40 - briefly describe to readers your rationale to reach your final formula
L46 - 48 - the cost of running was independent of speed under 20km/h. Where this affirmation comes from? If Dlim and parameter "a" were equivalent to amounts of energy, then are you implying that the speed for the mentioned events is under 20km/h?
L59 - 65 - please add references.
L71 - 73 - Nice work adding the references. For the sake of curiosity, it would be interesting to provide a brief description of how these studies came to find the exponent "g" values (0.95 and 0.80).
L91 - you only mention one study.
L92 - L97 - this is better, but still needs to be more clear. During the single-visit tests, participants would perform the 3600m, 2400m, 1200m all in the same session, with a 30- or 60-minute recovery between tests? As these are constant distance protocols, the reliability of it is much higher than the constant speed protocol of the treadmill testing. What was the protocol used for the constant speed treadmill test? If there was no difference between both, and a constant speed treadmill test would have a lower reliability, then are these field tests reliable?
Purpose and Methods
L105 - 107 - your purpose still seems confusing. "To confirm the interest of Scrit and Pcrit computed from exercise whose performances are submaximal". Isn't your purpose to determine Scrit and Pcrti from submaximal performances in the field?
You still have not provided a definition for low- and high-endurance athlete. Is there a specific criteria that differentiates both? Elite vs amateur status? VO2max? Specific performances?
The same goes for short and long performances.
Please explain the arbitrary units of 1-4 and 1-7, considering the Tlim is similar for both (3-12min and 2-15min).
Results and Discussion
This section is still confusing. You mention that the effects of submaximal performance on Scrit and Pcrit are low, and that these could explain that it is possible to compute Scrit from 3 submaximal tests with only a 30-minute recovery. But you also mention that there is low-sensitivity to it? Then is this a reliable method or not?
Again, the time limit that is more reliable (4-30min) is still incredibly large. Can you provide recommendations for the duration of the tests? Can you provide references that support a narrower range?
You added information on 30- and 60-minute recovery between the tests but never made it clear in the previous sections of the paper. Why these specific times? Are the submaximal tests still with the same distances previously mentioned? That would minimize fatigue and increase the reliability of the tests, but any other rationale?
The purpose is still not clear in the discussion. The idea was to show how Scrit and Pcrit can be calculated from these submaximal performances, but these don't seem to be discussed in detail so that it can be applied in practical settings. Which submaximal tests would be used? Again, see notes above on the duration of Tlim.
In short, this section could have been discussed in much more depth to elucidate how the findings of this study relate to what has been previously reported in the literature and to describe the novel findings of this stud (describing these in detail, so that it can be used in the future).
Conclusion
You mention that the reliability of protocol 3 would be better for trained athletes due to their ability to better pace themselves, but this is never discussed in the previous section. The conclusion has been enhanced, but now it seems that some of your statements here have not been discussed in depth in your discussion.
Author Response
Please modify the abstract according to the changes that were made and make it clearer when arriving at the conclusions. DONE
Introduction
Thank you for your review
L38 - 40 - briefly describe to readers your rationale to reach your final formula DONE
L46 - 48 - the cost of running was independent of speed under 20km/h.
22 km/h instead 20 km/hg
Where this affirmation comes from? Margaria et al. 1963 [8]
If Dlim and parameter "a" were equivalent to amounts of energy, then are you implying that the speed for the mentioned events is under 20km/h?
L59 - 65 - please add references. DONE
L71 - 73 - Nice work adding the references. For the sake of curiosity, it would be interesting to provide a brief description of how these studies came to find the exponent "g" values (0.95 and 0.80). DONE
L91 - you only mention one study. YES because the other studies were not related to SCrit or PCrit
L92 - L97 - this is better, but still needs to be more clear. During the single-visit tests, participants would perform the 3600m, 2400m, 1200m all in the same session, with a 30- or 60-minute recovery between tests? In two sessions
As these are constant distance protocols, the reliability of it is much higher than the constant speed protocol of the treadmill testing.NO
What was the protocol used for the constant speed treadmill test? The constant-speed runs to exhaustion on treadmill were performed with 3 running speeds during 3 separate sessions.
If there was no difference between both, and a constant speed treadmill test would have a lower reliability, then are these field tests reliable? YES
Purpose and Methods
L105 - 107 - your purpose still seems confusing. "To confirm the interest of Scrit and Pcrit computed from exercise whose performances are submaximal". Isn't your purpose to determine Scrit and Pcrti from submaximal performances in the field? Not only
You still have not provided a definition for low- and high-endurance athlete. Is there a specific criteria that differentiates both? Elite vs amateur status? VO2max? Specific performances? REFERENCE 31
The same goes for short and long performances.
Please explain the arbitrary units of 1-4 and 1-7, considering the Tlim is similar for both (3-12min and 2-15min).
Results and Discussion
This section is still confusing. You mention that the effects of submaximal performance on Scrit and Pcrit are low, and that these could explain that it is possible to compute Scrit from 3 submaximal tests with only a 30-minute recovery. But you also mention that there is low-sensitivity to it? Then is this a reliable method or not?
For SCrit or PCrit in the present theoretical study, the effects of 20%-submaximal performances in protocol 1 are lower than the effects of 5%-submaximal performances in protocols 2 and 3.
Again, the time limit that is more reliable (4-30min) is still incredibly large. Can you provide recommendations for the duration of the tests? Can you provide references that support a narrower range? NO
You added information on 30- and 60-minute recovery between the tests but never made it clear in the previous sections of the paper. Why these specific times? Are the submaximal tests still with the same distances previously mentioned? YES
The purpose is still not clear in the discussion. The idea was to show how Scrit and Pcrit can be calculated from these submaximal performances, but these don't seem to be discussed in detail so that it can be applied in practical settings. Which submaximal tests would be used? Again, see notes above on the duration of Tlim.
In short, this section could have been discussed in much more depth to elucidate how the findings of this study relate to what has been previously reported in the literature and to describe the novel findings of this stud (describing these in detail, so that it can be used in the future).
Conclusion
You mention that the reliability of protocol 3 would be better for trained athletes due to their ability to better pace themselves, but this is never discussed in the previous section. The conclusion has been enhanced, but now it seems that some of your statements here have not been discussed in depth in your discussion. IMPROVED
Reviewer 3 Report
I would like to thank the authores for thier response.
Author Response
Thank you for your review
The paper has been modified in response to reviewer 1